# Factors associated with psychological symptoms in hospital workers of a French hospital during the COVID-19 pandemic: Lessons from the first wave

M. d'Ussel[1]*, A. Fels[2], X. Durand[3], C. Lemogne[4‡], G. Chatellier[5‡], N. Castreau[6‡], F. Adam[7]

1 Consultation Douleur Chronique, Groupe Hospitalier Paris Saint-Joseph, Paris, France, 2 Centre de Recherche Clinique, Groupe Hospitalier Paris Saint-Joseph, Paris, France, 3 Service de Chirurgie Urologique, Groupe Hospitalier Paris Saint-Joseph, Paris, France, 4 AP-HP, Hôpital Hôtel-Dieu, DMU Psychiatrie et Addictologie, Service de Psychiatrie de l'Adulte, INSERM, Institut de Psychiatrie et Neurosciences de Paris (IPNP), UMR_S1266, Université de Paris, Paris, France, 5 Département de Recherche Clinique, Groupe Hospitalier Paris Saint-Joseph, Paris, France, 6 Service de Santé au Travail, Groupe Hospitalier Paris Saint-Joseph, Paris, France, 7 Département d'Anesthésie, Groupe Hospitalier Paris Saint-Joseph, Paris, France

☉ These authors contributed equally to this work.
‡ CL GC and NC also contributed equally to this work.
* mdussel@ghpsj.fr

**Data Availability Statement:** All relevant data are within the manuscript and its Supporting Information files (as an Excel file).

## Abstract

### Purpose

The COVID-19 pandemic has put hospital workers around the world in an unprecedented and difficult situation, possibly leading to emotional difficulties and mental health problems. We aimed to analyze psychological symptoms of the hospital employees of the Paris Saint-Joseph Hospital Group a few months after the 1st wave of the pandemic.

### Participants and methods

From July 15 to October 1, 2020, a cross-sectional survey was conducted among hospital workers in the two locations of our hospital group using the Hospital Anxiety and Depression Scale (HADS) and Post-Traumatic Stress Disorder (PTSD) Checklist (PCL) to measure anxiety, depression, and PTSD symptoms. Factors independently associated with these symptoms were identified.

### Results

In total, 780 participants (47% caregivers, 18% health administrative workers, 16% physicians, and 19% other professionals) completed the survey. Significant symptoms of anxiety, depression, and PTSD were reported by 41%, 21%, and 14% of the participants, respectively. Hierarchical regression analysis showed a higher risk of having psychological symptoms among those (1) who were infected by SARS-CoV-2 or had colleagues or relatives infected by the virus, (2) who retrospectively reported to have had an anxious experience during the first wave, and (3) with a previous history of burnout or depression. In contrast,

**Funding:** The author(s) received no specific funding for this work.

**Competing interests:** I have read the journal's policy and the authors of this manuscript have the following competing interests: Marguerite d'Ussel reports personal fees and non-financial support from Grunenthal, Ethypharm and IPRAD in the previous three years, outside the submitted work. Xavier Durand reports personal fees and non-financial support from JANSSEN, ASTELLAS, RECORDATI and IPSEN in the previous three years, outside the submitted work. Cédric Lemogne reports personal fees and non-financial support from Janssen-Cilag, Lundbeck, Otsuka Pharmaceutical, and Boehringer Ingelheim in the previous three years, outside the submitted work.

job satisfaction appeared to be a protective factor. Overall, hospital workers showed the statistically same levels of anxiety, depression, and PTSD symptoms, regardless of their profession and whether they had worked in units with COVID-19 patients or not.

## Conclusions

Our cross-sectional survey of 780 hospital workers shows that after the first wave, hospital employees had a significant burden of mental health symptoms. Specific preventive measures to promote mental well-being among hospital workers exposed to COVID-19 need to be implemented, first among particularly vulnerable staff, and then, for all hospital staff for whom anxiety is detected early, and not only those who were directly exposed to infected patients.

## Introduction

On January 20, 2020, the World Health Organization (WHO) declared the disease caused by the novel coronavirus, severe acute respiratory syndrome coronavirus 2 (SARS-CoV2), called coronavirus disease 2019 (COVID-19), to be a public health emergency. Soon afterwards, as COVID-19 started to show the characteristics of rapid and wide transmission, the WHO reclassified the problem as a pandemic. With more than six million confirmed cases worldwide and more than 350,000 deaths declared between February and May 2020, this pandemic emerged as an unprecedented healthcare crisis.

Previous research has shown that, historically, exposure to pandemics is accompanied by mental health problems in the general population and healthcare workers, such as symptoms of post-traumatic stress disorder (PTSD), anxiety, and depression [1]. Such psychiatric symptoms among healthcare workers have been reported during the outbreak, but also away from the epidemic [2, 3].

WHO interim guidance from March 2020 [4] emphasized that the COVID-19 pandemic inevitably places healthcare workers at risk. While healthcare professionals must cope with the same societal shifts and emotional stressors faced by all people, they also face a greater risk of exposure, extreme workloads, moral dilemmas, and a rapidly evolving practice environment that differs greatly from that with which they are familiar.

In a meta-analysis of 13 studies (33,062 participants) evaluating the mental health of healthcare workers during the COVID-19 pandemic, the prevalence rate of anxiety and depression symptoms was 23.2% and 22.8%, respectively [5]. Unlike certain countries, France has not had to deal with other recent epidemics, such as SARS-1, Middle East Respiratory Syndrome, or Ebola. A French study, carried out among critical care clinicians managing patients with COVID-19 during the first wave, estimated that the prevalence of anxiety, depression, and peri-traumatic dissociation symptoms was 50.4%, 30.4%, and 32.0%, respectively [6].

Many hospital employees shared risk factors with healthcare workers because of their direct exposure to infected patients, the limited availability of protective equipment, and the increased workload related to the pandemic. It is, thus, likely that non-clinical support staff, such as administrative assistants, security, and environmental services personnel, were also subject to the same effects on their mental health. Specific literature on this subject is, however, scarce.

Our hospital group (Groupe Hospitalier Paris Saint-Joseph) is composed of two hospitals (Hospital Paris Saint-Joseph and Hospital Marie-Lannelongue). During the first wave (between March 1 and May 29, 2020) 1,177 patients were managed in the emergency ward, 834 were hospitalized, 132 needed intensive care, and 100 died.

Here, we aimed to determine the risk factors associated with mental health symptoms (anxiety, depression, and PTSD) after the first wave of the COVID-19 pandemic among all components of the workforce in our hospital group. The findings of such a study could be informative for decision making during future waves of the pandemic and potential pandemics caused by other agents that may follow.

## Methods

The study sponsor was the "Groupe Hospitalier Paris Saint-Joseph". The protocol was performed in accordance with the Declaration of Helsinki and approved by the institutional ethics committee: the GERM (Groupe Ethique et Recherche Médicale) from the Hospital Paris Saint-Joseph (IRB number IRB00012157 -n° initial agreement 436 and registered on the national institute of health data platform).

No written informed consent was required. The authors guarantee the anonymization of all data collected.

The study started on July 15, 2020, three months after the peak of the pandemic in France, and finished on October 1, 2020. All hospital workers were invited to complete an online survey, with a link sent to the mailing list. The use of a "Microsoft Forms" electronic format allowed us to securely send the survey and store the answers. Informed consent was obtained from all participants on the first page of the survey.

To be included, the only criteria was to have worked in the hospital between March 15 and May 15, 2020.

The survey included five items that were identified from a literature review [3, 7–9] and from interviews with Groupe Hospitalier Paris Saint-Joseph workers. These items were: (the entire questionnaire can be found in the S1 Appendix)

1. Individual characteristics (demographics, exposure to COVID-19, history of burnout)

2. Professional experience and fears at the time of the survey

3. Emotional experience during the first wave and the degree of job satisfaction:

   a. Did the COVID-19 crisis make you anxious? If so: for you, your family, others, or in your work?

   b. Were you afraid of infecting your loved ones?

   c. Currently, do you feel good at work?

4. The Hospital Anxiety and Depression Scale (HADS). The HADS is a 14-item auto questionnaire that includes seven items about symptoms of anxiety and seven items on symptoms of depression [10]. A cutoff score > 7 was used for each subscale to detect significant symptoms of anxiety or depression [11, 12].

5. The PTSD Checklist (PCL). The PCL is a 17-item self-reported measure reflecting DSM-IV symptoms of PTSD [13]. A cutoff score > 44 was used to detect significant symptoms of PTSD [14]. It was made clear in the questionnaire that the stressful event to which the respondents had to refer was the health crisis. At the time of the survey, it was thought that the pandemic was over and had been over for more than a month.

## Statistical analysis

Data are described as numbers or percentages. Variables were compared using either χ2-test or Fisher exact tests, as appropriate.

Independent predictors for anxiety, depression, and PTSD symptoms were assessed using logistic regression. Odds ratios (ORs) and their 95% confidence intervals (95% CIs) were calculated using univariate and multivariate logistic regression models. We conducted several logistic regressions. We first calculated the unadjusted models and then conducted multiple logistic regression analyses using a hierarchical approach. The first model included socio-demographic variables and the second the same variables plus variables related to the emotional experience during the first wave of the COVID-19 pandemic (personal experience of anxiety, fear of infecting loved ones, fear of being infected by SARS-CoV-2, and job satisfaction). We also assessed potential interactions.

Variables of interest were selected according to their statistical significance in univariate analysis (critical p-value for entry into the model < 0.1) or were forced into the model (age and sex).

Analyses were performed using R software (the R project for statistical computing, https:// www.r-project.org/). All tests were two-sided and a p-value < 0.05 was considered statistically significant.

## Results

### Participant characteristics (Table 1)

Among the 3,272 employees working in the two locations of our hospital group, 780 participated in the survey. Among them, 1,675 worked at the Hospital Paris Saint-Joseph (which has 2,408 employees), and 105 at the Hospital Marie-Lannelongue (which has 864 employees) (Table 1).

The largest proportion (47%) of healthcare staff were caregivers (nurses, assistant nurses, nurse managers), followed by 18% healthcare administrators (secretaries, logistic managers, pharmaceutical assistants), and 16% medical doctors (physicians, pharmacists, or medical technicians).

Among the participants, 56% had been in the same workplace for less than five years. A previous history of burnout or depression was reported by 22%. Forty-eight percent reported having worked in a unit with COVID-19 patients during the first wave, 44% in another clinical unit, and the remaining participants were distributed among different types of units.

Almost half of the participants (46%) reported having frequently worked with COVID-19 patients (every working day) or regularly (at least once a week). Nearly 60% reported that a colleague, friend, or close relative had a SARS-CoV-2 infection.

### Emotional experience during the COVID-19 first wave

Among the respondents, 62% reported that they were anxious during the period of the first wave; most (86%) were anxious for their family, 57% for themselves, 51% at work, and 42% for others. Participants were more afraid of infecting their loved ones (75%) than being infected themselves by SARS-CoV-2 (45%).

Three quarters of hospital workers indicted they had high job satisfaction at the moment of the survey.

### HADS and PCL results

Clinically significant levels of anxiety symptoms (HADS-Anxiety = HADa > 7) were found for 41% of the respondents, depression symptoms (HADS-Depression = HADd > 7) for 21%, and PTSD symptoms (PCL > 44) for 14%.

**Table 1. Respondents' characteristics and reports about their COVID-19 experience.**

| Variable (N = 780) | N (%) |
|---|---|
| **Gender** (N = 780) | |
| • Women | 638 (81.8%) |
| • Men | 142 (18.2%) |
| **Age** (years) (N = 780) | |
| • ≤ 41 | 429 (55.0%) |
| • > 41 | 351 (45.0%) |
| **Marital situation** (N = 747) | |
| • Single | 235 (31.5%) |
| • Living as a couple | 512 (68.5%) |
| **Familial situation** (N = 471) | |
| • No children | 72 (15.3%) |
| • One or several children | 399 (84.7%) |
| **Location of professional practice** (N = 780) | |
| • Hospital Paris Saint-Joseph | 675 (86.5%) |
| • Hospital Marie-Lannelongue | 105 (13.5%) |
| **Profession** (N = 780) | |
| • Caregiver (nurse, assistant nurse, nurse manager) | 367 (47.1%) |
| • Healthcare administrator | 141 (18.1%) |
| • Medical professional (physician, pharmacologist, biologist) | 125 (16.0%) |
| • Other caregivers (physiotherapist, stretcher-bearer, radiological technologist, psychologist) | 72 (9.2%) |
| • Midwife | 21 (2.7%) |
| • Others | 54 (6.9%) |
| **Professional experience in the same hospital unit** (years) (N = 780) | |
| < 5 | 437 (56.0%) |
| ≥ 5 | 343 (44.0%) |
| **History of professional burnout or depression** (N = 780) | |
| No | 605 (77.6%) |
| Yes | 175 (22.4%) |
| **Duty station during the COVID-19 crisis** (N = 780) | |
| • COVID Unit | 375 (48.1%) |
| • Non-Covid Unit | 347 (44.5%) |
| • Other professional activity | 143 (18.3%) |
| • Remote work | 105 (13.5%) |
| **COVID-19 patient management** (N = 780) | |
| • Frequently / Regularly | 361 (46.3%) |
| • Rarely / Never | 419 (53.7%) |
| **Respondents who were infected or with colleagues or relatives who were infected** (N = 780) | |
| Yes | 466 (59.7%) |
| No | 314 (40.3%) |

In unadjusted models, several factors were significantly associated with the odds of probable anxiety, depression, and PTSD symptoms to varying extents (Tables 2–4 and Table A-C in S1 Table). There were fewer men in the group with symptoms of anxiety than in the group without (13% vs 22%, p = 0.001), as well as in the group with symptoms of PTSD than that without (11% vs 19%, p = 0.029). However, there were as many men in the group of workers

**Table 2. Unadjusted and adjusted logistic regression for symptoms of anxiety (HADa > 7), n = 780.**

| | Unadjusted | | Model 1* | | Model 2** | |
|---|---|---|---|---|---|---|
| | OR [95%CI] | p | OR [95%CI] | p | OR [95%CI] | p |
| **Gender:** Women (vs Men) | 1.97 [1.32;2.93] | 0.001 | 1.70 [1.11;2.64] | 0.015 | 1.48 [0.90;2.45] | 0.12 |
| Age (years) ≤ 41 (vs > 41) | 0.90 [0.67;1.19] | 0.45 | 0.89 [0.65;1.20] | 0.44 | 0.94 [0.66;1.33] | 0.72 |
| **Marital situation:** Living as a couple (vs single) | 0.78 [0.57;1.07] | 0.12 | | | | |
| **Familial situation:** One or several children vs no children | 1.01 [0.61;1.68] | 0.98 | | | | |
| **Profession** (vs Healthcare administrator) | | 0.033 | | | | |
| • Medical professional | 0.45 [0.27;0.76] | | 0.56 [0.32;0.96] | 0.035 | 0.54 [0.29;1.00] | 0.05 |
| • Caregiver | 0.88 [0.60;1.31] | | 0.87 [0.58;1.31] | 0.50 | 0.76 [0.47;1.21] | 0.24 |
| • Other Caregiver | 0.74 [0.42;1.33] | | 0.75 [0.41;1.37] | 0.36 | 0.97 [0.48;1.93] | 0.92 |
| • Midwife | 0.58 [0.22;1.54] | | 0.57 [0.20;1.51] | 0.27 | 0.40 [0.12;1.24] | 0.12 |
| • Other | 1.01 [0.54;1.89] | | 1.15 [0.60;2.21] | 0.67 | 1.35 [0.63;2.89] | 0.43 |
| **Location of professional practice:** Hospital Marie-Lannelongue (vs Hospital Paris Saint-Joseph) | 0.96 [0.63;1.47] | 0.86 | 0.96 [0.61;1.49] | 0.85 | 0.81 [0.48;1.34] | 0.41 |
| **Professional experience in the same hospital area:** > 5 years vs < 5 years | 0.94 [0.71;1.25] | 0.68 | | | | |
| **Duty station during the COVID-19 crisis** | | | | | | |
| COVID-area assignment | 0.79 [0.60;1.06] | 0.11 | | | | |
| Non COVID-area assignment | 0.97 [0.73;1.29] | 0.83 | | | | |
| Remote work | 1.38 [0.91;2.09] | 0.12 | | | | |
| Non-clinical professional activity | 1.03 [0.71;1.48] | 0.89 | | | | |
| **COVID-19 patient management:** regularly / frequently vs never/rarely | 0.79 [0.59;1.05] | 0.102 | | | | |
| **Respondents who were infected or with colleagues or relatives who were infected:** yes vs no | 1.75 [1.30;2.36] | < 0.001 | 1.63 [1.19;2.23] | 0.002 | 1.55 [1.09;2.23] | 0.016 |
| **Anxiety during the 1st Wave:** yes vs no | 6.85 [4.77;9.84] | < 0.001 | | | 5.83 [3.89;8.90] | < 0.001 |
| • **Anxiety for oneself:** yes vs no | 1.18 [0.82;1.69] | 0.37 | | | | |
| • **Anxiety for family:** yes vs no | 1.03 [0.61;1.74] | 0.92 | | | | |
| • **Anxiety for others:** yes vs no | 0.92 [0.64;1.33] | 0.67 | | | | |
| • **Anxiety at work:** yes vs no | 1.43 [1.00;2.05] | 0.049 | | | | |
| **Fear of contaminating relatives during the 1st wave:** yes vs no | 2.50 [1.75;3.58] | < 0.001 | | | 1.64 [1.04;2.59] | 0.034 |
| **History of professional burnout or depression:** yes vs no | 2.94 [2.08;4.16] | < 0.001 | 2.79 [1.96;4.00] | < 0.001 | 2.63 [1.75;3.97] | < 0.001 |
| **Date of history of depression or burnout (N = 175):** > 3 years vs < 3 years | 0.53 [0.28;1.00] | 0.05 | | | | |

*(Continued)*

**Table 2.** (Continued)

| | Unadjusted | | Model 1* | | Model 2** | |
|---|---|---|---|---|---|---|
| | OR [95%CI] | p | OR [95%CI] | p | OR [95%CI] | p |
| **Current job satisfaction:** yes vs no | 0.20 [0.14;0.28] | < 0.001 | | | 0.19 [0.13;0.29] | < 0.001 |

Model 1 included socio-demographic variables and model 2 the same variables plus variables related to the emotional experience during the 1st wave of the COVID-19 pandemic (personal experience of anxiety during the 1st wave due to fear of infecting loved ones or being infected by SARS-CoV-2 and job satisfaction).

HADa = HAD-Anxiety.

*Nagelkerke Pseudo-$R^2$: 0.11.

**Nagelkerke Pseudo-$R^2$: 0.37.

with symptoms of depression as in the group without (17% vs 19%, p = 0.73). There were fewer workers who were married or partnered in the group with symptoms of PTSD than in the group without (56% vs 71%, p = 0.003). In the group with symptoms of anxiety, there were fewer medical professionals than in the group without (11% vs 20%, p = 0.033). There were fewer healthcare workers working in a COVID-19 area in the group with symptoms of depression than in the group without (40% vs 50%, p = 0.023). There were more employees who were infected or had colleagues or relatives who were infected with SARS-CoV-2 in the group with symptoms of anxiety than in the group without (68% vs 54%, p < 0.001), as well as in the group with symptoms of PTSD relative to the group without (74% vs 57%, p < 0.001).

In the group with symptoms of anxiety, more hospital workers declared that they were anxious during the first wave of the pandemic than in the group without symptoms of anxiety (86% vs 46%, p = 0.001); the same was true for symptoms of depression (76% vs 59%, p < 0.001) and PTSD (90% vs 58%, p < 0.001). In addition more employees reported a history of burnout or depression in the group with symptoms of anxiety than in the group without (34% vs 15%, p < 0.001); the same was true for symptoms of depression, (39% vs 18%, p < 0.001) and PTSD (34% vs 21%, p = 0.001). Finally, there were fewer employees who had high job satisfaction in the group with symptoms of anxiety than in the group without (57% vs 87%, p < 0.001) and the same results were found for symptoms of depression (46% vs 82%, p < 0.001) and PTSD (48% vs 79%, p < 0.001).

## Multivariable analysis

The relationship between the female gender and anxiety remained significant, although smaller in magnitude, after adjusting for demographic characteristics, working conditions, and a history of professional burnout or depression (OR, 1.70; 95% CI: 1.11–2.64) (Table 2, model 1). However, the association between gender and anxiety (Table 2, model 2) was no longer significant after adjusting for the emotional experience of the first wave (OR, 1.48; 95% CI: 0.90–2.45), suggesting that this variable may substantially explain the association.

We observed a comparable pattern for the relationship between the medical profession and anxiety, which remained significant (Table 2, model 1) after adjusting for demographic characteristics, working conditions, and a history of professional burnout or depression (OR, 0.56; 95% CI: 0.32–0.96). However, adjusting for the emotional experience of the first wave made this relationship non-significant (OR, 0.54; 95% CI: 0.29–1.00).

Finally, the relationship between a history of professional burnout or depression and PTSD remained significant after adjusting for demographic characteristics and the working condition (OR, 1.80; 95% CI: 1.13–2.83) (Table 4, model 1) but not after adjusting for the emotional experience of the first wave (OR, 1.47; 95% CI: 0.89–2.39) (Table 4, model 2).

**Table 3. Unadjusted and adjusted logistic regression for symptoms of depression (HADd > 7), n = 780.**

| | Unadjusted | | Model 1* | | Model 2** | |
|---|---|---|---|---|---|---|
| | OR [95%CI] | p | OR [95%CI] | P | OR [95%CI] | p |
| **Gender**: Women (vs Men) | 1.08 [0.69;1.71] | 0.73 | 1.04 [0.65;1.71] | 0.86 | 0.88 [0.53;1.50] | 0.64 |
| **Age (years)** ≤ 41 (vs > 41) | 1.14 [0.80;1.61] | 0.47 | 1.09 [0.75;1.59] | 0.65 | 1.24 [0.83;1.85] | 0.30 |
| **Marital situation**: Living as a couple (vs single) | 0.70 [0.48;1.01] | 0.05 | 0.72 [0.49;1.06] | 0.09 | 0.75 [0.50;1.13] | 0.16 |
| **Familial situation**: One or several children vs no children | 1.21 [0.63;2.31] | 0.56 | | | | |
| **Profession** (vs healthcare administrator) | | 0.17 | | | | |
| • Medical professional | 0.64 [0.36;1.15] | | | | | |
| • Caregiver | 0.61 [0.38;0.96] | | | | | |
| • Other Caregiver | 0.97 [0.51;1.85] | | | | | |
| • Midwife | 0.29 [0.06;1.28] | | | | | |
| • Other | 0.77 [0.37;1.63] | | | | | |
| **Location of professional practice**: Hospital Marie-Lannelongue (vs Hospital Paris Saint-Joseph) | 1.38 [0.86;2.23] | 0.20 | 1.40 [0.82;2.32] | 0.20 | 1.22 [0.69;2.12] | 0.48 |
| **Professional experience in the same hospital area**: > 5 years vs < 5 years | 1.16 [0.82;1.64] | 0.40 | | | | |
| **Duty station during COVID-19 crisis** | | | | | | |
| • COVID-area assignment: yes vs no | 0.67 [0.47;0.95] | 0.023 | 0.72 [0.49;1.07] | 0.10 | 0.71 [0.47;1.07] | 0.10 |
| • Non COVID-area assignment: yes vs no | 1.24 [0.88;1.76] | 0.22 | | | | |
| • Remote work: yes vs no | 1.65 [1.03;2.62] | 0.034 | 1.54 [0.92;2.54] | 0.09 | 1.41 [0.81;2.42] | 0.22 |
| • Non-clinical professional activity: yes vs no | 0.91 [0.58;1.44] | 0.70 | | | | |
| **COVID-19 patient management**: regularly / frequently vs never/rarely | 0.88 [0.62;1.25] | 0.48 | | | | |
| **Respondents who were infected or with colleagues or relatives who were infected** (yes vs no) | 1.31 [0.91;1.88] | 0.14 | 1.33 [0.91;1.96] | 0.14 | 1.22 [0.81;1.83] | 0.35 |
| **Anxiety during 1st Wave**: yes vs no | 2.22 [1.49;3.29] | <0.001 | | | 1.70 [1.11;2.65] | 0.017 |
| • **Anxiety for oneself:** yes vs no | 1.11 [0.73;1.69] | 0.61 | | | | |
| • **Anxiety for family:** yes vs no | 1.04 [0.57;1.91] | 0.89 | | | | |
| • **Anxiety for others:** yes vs no | 0.95 [0.63;1.44] | 0.81 | | | | |
| • **Anxiety at work:** yes vs no | 0.98 [0.65;1.47] | 0.91 | | | | |
| **History of professional burnout or depression:** yes vs no | 2.88 [1.97;4.19] | < 0.001 | 2.53 [1.70;3.74] | <0.001 | 2.25 [1.47;3.43] | < 0.001 |
| **Current job satisfaction:** yes vs no | 0.18 [0.12;0.26] | < 0.001 | | | 0.20 [0.13;0.29] | < 0.001 |

Model 1 included socio-demographic variables and model 2 the same variables plus variables related to the emotional experience during the 1st wave of the COVID-19 pandemic (personal experience of anxiety during the 1st wave due to fear of infecting loved ones or of being infected by SARS-CoV-2 and job satisfaction).

HADd = HAD-Depression.

*Nagelkerke Pseudo-$R^2$: 0.14.

**Nagelkerke Pseudo-$R^2$: 0.27.

**Table 4. Unadjusted and adjusted logistic regression for symptoms of post-traumatic stress (PCL > 44), n = 780.**

| | Unadjusted | | Model 1* | | Model 2** | |
|---|---|---|---|---|---|---|
| | OR [95%CI] | p | OR [95%CI] | p | OR [95%CI] | p |
| **Gender**: Women (vs Men) | 1.99 [1.06;3.73] | 0.029 | 1.68 [0.91;3.35] | 0.12 | 1.33 [0.69;2.77] | 0.41 |
| **Age (years)** ≤ 41 (vs > 41) | 0.77 [0.51;1.17] | 0.22 | 0.85 [0.55;1.31] | 0.46 | 0.83 [0.52;1.31] | 0.43 |
| **Marital situation**: Living as a couple (vs single) | 0.53 [0.35;0.81] | 0.003 | 0.55 [0.35;0.84] | 0.006 | 0.50 [0.32;0.80] | 0.003 |
| **Familial situation**: One or several children vs no children | 0.97 [0.47;2.01] | 0.94 | | | | |
| **Profession** (vs healthcare administrator) | | 0.12 | | | | |
| Medical professional | 0.47 [0.21;1.04] | | | | | |
| Caregiver | 1.10 [0.65;1.87] | | | | | |
| Other Caregiver | 0.58 [0.24;1.44] | | | | | |
| Midwife | 1.27 [0.39;4.14] | | | | | |
| Other | 0.68 [0.26;1.77] | | | | | |
| **Location of professional practice**: Hospital Marie-Lannelongue (vs Hospital Paris Saint-Joseph) | 1.19 [0.68;2.10] | 0.55 | 1.41 [0.74;2.55] | 0.27 | 1.29 [0.66;2.43] | 0.44 |
| **Professional experience in the same hospital unit**: > 5 years vs < 5 years | 1.01 [0.67;1.51] | 0.97 | | | | |
| **Duty station during the COVID-19 crisis** | | | | | | |
| COVID-area assignment | 1.17 [0.78;1.74] | 0.46 | | | | |
| Non COVID-area assignment | 1.07 [0.72;1.60] | 0.74 | | | | |
| Remote work | 0.92 [0.50;1.67] | 0.78 | | | | |
| Non-clinical professional activity | 0.55 [0.30;1.01] | 0.05 | 0.62 [0.32;1.14] | 0.14 | 0.73 [0.36;1.39] | 0.36 |
| **COVID-19 patients management**: Regularly / Frequently vs never/rarely | 1.03 [0.69;1.54] | 0.90 | | | | |
| **Respondents who were infected or with colleagues or relatives who were infected** (yes vs no) | 2.10 [1.34;3.29] | 0.001 | 2.10 [1.33;3.41] | 0.002 | 1.99 [1.22;3.33] | 0.007 |
| **Anxiety during the 1st Wave**: yes vs no | 6.67 [3.51;12.66] | <0.001 | | | 4.47 [2.35;9.27] | < 0.001 |
| • **Anxiety for oneself**: yes vs no | 1.27 [0.81;1.99] | 0.30 | | | | |
| • **Anxiety for family**: yes vs no | 1.32 [0.66;2.62] | 0.43 | | | | |
| • **Anxiety for others**: yes vs no | 1.14 [0.73;1.78] | 0.55 | | | | |
| • **Anxiety at work**: yes vs no | 1.87 [1.19;2.94] | 0.006 | | | | |
| **Fear of contaminating relatives during the 1st wave**: yes vs no | 3.15 [1.69;5.88] | < 0.001 | | | 2.52 [1.26;5.55] | 0.014 |
| **History of professional burnout or depression**: yes vs no | 2.02 [1.31;3.12] | 0.001 | 1.80 [1.13;2.83] | 0.011 | 1.47 [0.89;2.39] | 0.12 |
| **Date of history of depression or burnout (N = 175)**: > 3years vs < 3 years | 0.68 [0.33;1.40] | 0.29 | | | | |
| **Current job satisfaction**: yes vs no | 0.24 [0.16;0.36] | < 0.001 | | | 0.28 [0.18;0.44] | < 0.001 |

Model 1 included socio-demographic variables and model 2 the same variables plus variables related to the emotional experience during the 1st wave of the COVID-19 pandemic (personal experience of anxiety during the 1st wave due to fear of infecting loved ones or of being infected by SARS-CoV-2 and job satisfaction).

*Nagelkerke Pseudo-$R^2$: 0.13.

**Nagelkerke Pseudo-$R^2$: 0.28.

In multivariable analysis (model 2), healthcare workers who retrospectively reported that they felt anxious during the first wave showed more symptoms of anxiety (OR, 5.83; 95% CI: 3.89–8.90), depression (OR, 1.70; 95% CI: 1.11–2.65), and PTSD (OR, 4.47; 95% CI: 2.35–9.27) at the moment of the survey. Hospital workers with a history of burnout or depression had an

increased risk of anxiety (OR, 2.63; 95% CI: 1.75–3.97) and depression (OR, 2.25; 95% CI: 1.47–3.43).

The single variable associated with a lower risk of all three studied mental health symptoms (model 2) was high job satisfaction (OR, 0.19; 95% CI: 0.13–0.29 for anxiety; OR, 0.20; 95% CI: 0.13–0.29 for depression; and OR, 0.28; 95% CI, 0.18–0.44 for PTSD).

Being married or partnered was independently associated with a lower risk of PTSD symptoms (Table 4, model 2) (OR, 0.50; 95% CI: 0.32–0.80).

## Discussion

Our cross-sectional survey, which included 780 hospital workers, identified a prevalence of probable anxiety, depression, and PTSD after the first wave of the COVID-19 pandemic of 41%, 21%, and 14%, respectively. The following three factors were independently associated with the presence of the three types of psychological symptoms: those who reported to have been anxious during the first wave, those with a previous history of burnout or depression, and those who had themselves been infected or who had colleagues or relatives who were infected. This suggests that at the beginning of any epidemic emergency, screening for ongoing psychological problems among healthcare workers should be carefully conducted by hospital management to protect those who are the most vulnerable as a primary preventive measure.

Our survey also showed that high job satisfaction had a positive influence, resulting in better mental health scores.

Other studies in the literature concerning healthcare workers during the COVID-19 pandemic reported a prevalence of anxiety from 23% to 51% and depression from 22% to 50% [15–17] based on assessment measurements. A systematic review of 29 studies reported a median prevalence of 24% for anxiety and 21% for depression among healthcare workers [18].

Our hospital workers had a higher prevalence of anxiety symptoms than the general French population over the same period but a lower frequency of depressive symptoms based on results published on the "Santé Publique France" website [19]. In an August 2020 survey, approximately 18% of the French population interviewed reported anxiety (21% for our hospital workers for the same HAD scale cut-off as in the survey) and 11% depression (9% for our hospital workers).

The prevalence of COVID-19 related-PTSD symptoms among healthcare workers has been reported to be 71% [20], 21% [21], and 15% [22] versus our rate of 14%. This difference in the prevalence is likely linked to the instruments we used to assess symptoms and the moment when the survey was carried out: during the peak of the pandemic versus after the end of the first wave. Indeed, several studies carried out after the SARS and MERs epidemics found significant rates of PTSD several months after the epidemic [2, 23]. Our survey was conducted from 2 to 6 months after the first wave.

We found no difference in the prevalence of psychological symptoms between administrative staff and healthcare workers or between staff working in zones with COVID-19 patients and those without. Employees working in the two hospitals showed the same levels of emotional suffering after the first wave, whether they were directly exposed to patients or not. Many previous studies have reported a higher prevalence of emotional symptoms among front-line healthcare workers [24, 25]. However, the results of several recent studies are in accordance with ours, with an identical emotional impact on healthcare workers directly exposed to COVID-19 patients land those who are not [23, 26, 27]. Liang et al. [28] even showed that frontline nurses working with COVID-19 patients had significantly lower traumatization scores than non-frontline nurses, the general public, or medical teams aiding COVID-19 control efforts. Indeed, worries related to the fear of being infected or higher-than-usual

exposure to death among COVID-19 units might have been balanced by the higher availability of protection equipment and greater feeling of usefulness. As the impact of constraints on mental health can be offset by higher rewards [28], the societal praise of healthcare professionals working in COVID-19 units may have acted as a protective factor, much like the level of job satisfaction. Another explanation may be that the anxiety caused by the sudden population lock-down was more dominant than factors intrinsic to their hospital workplace, as suggested by Milgrom et al [26].

One factor that was independently associated with higher symptoms of anxiety, depression, and PTSD was to have been anxious during the first wave. We certainly cannot draw conclusions about the relationship between anxiety during the event and the occurrence of distant emotional symptoms based on the results of a single study using such a retrospective measure by a dichotomous question. In addition, such a relationship could have resulted from the simple fact that people with a higher level of general anxiety are prone to have been more anxious in the past. However, this observation could lead to a prospective study to evaluate the impact of secondary prevention strategies targeting workers who feel the most anxious during a crisis.

Another factor associated with a significantly increased risk of showing adverse psychological outcomes was having been infected by the virus, or having colleagues or relatives who had been infected. This factor has already been found in other studies [25, 29] and suggests that this population would be more vulnerable to emotional distress and should receive specific support. A high risk of infection may leave workers feeling vulnerable because COVID-19 is highly infectious, has a high morbidity rate, and is potentially fatal [28].

We observed that employees who had already had depression or a history of burnout had more symptoms of anxiety and depression. This result is not surprising. It has been shown that people with a pre-existing mental health disease were among the groups at highest risk for a range of psychiatric distress symptoms during the COVID-19 pandemic [30]. Moreover, those with anxiety-related disorders reported greater fears about danger and contamination and traumatic stress symptoms [31].

On the contrary, we observed a lower prevalence for the three studied mental health disorders among respondents who reported high job satisfaction. Accordingly, Wang et al. showed that job satisfaction was a factor related to PTSD of nurses exposed to COVID-19 in China [32]. Mental health is highly relevant to work satisfaction. Healthcare workers who are dissatisfied with their job often feel that they are working in a dysfunctional system that affects the quality of their tasks and their self-esteem, factors associated with a higher risk of developing symptoms of anxiety, depression, and stress [20].

We used a hierarchical strategy in the multivariable analysis to observe the impact of the emotional experience during the crisis on the studied variables: the risk of symptoms of anxiety was higher for women in model 1, but by adding the emotional experience as a variable in model 2, anxiety and female gender were no longer significantly associated. Thus, although female gender was strongly associated with symptoms of anxiety, the causal pathway for this relationship was likely to have been through the emotional experience during the crisis.

We found similar results for participants with a history of burnout or depression, who had more PTSD symptoms. When we adjusted for the variable of emotional experience during the crisis, a history of burnout or depression was no longer associated with PSTD symptoms. The relationship between a history of burnout or depression and PTSD also appears to have been mediated by the emotional experience of the crisis.

The impact of the personal experience of anxiety during the crisis was particularly strong when the respondents reported that their anxiety was focused on their job. There appears to be a strong relationship between such anxiety just at the time of the first wave (reported retrospectively by the respondents) and the prevalence of significant symptoms of anxiety,

depression, and PTSD afterwards. Thus, by identifying those employees who feel the most anxious during a health crisis it may be possible to implement secondary prevention strategies targeted to these staff members in the hope of reducing the prevalence of post-crisis mental health disorders.

In our study, medical professionals reported less anxiety. Resilience, which is defined as the capacity to cope with and positively adapt to adversity, is an important protective factor and is of particular concern to researchers in the field of adversity [28]. It has been demonstrated during the COVID-19 epidemic that resilience can help to reduce worry, anxiety, and depression [33, 34]. Medical staff members have a higher level of education, which is positively related to resilience [35]. They also generally have greater decisional latitude at work, which is a protective factor against professional stress.

Marital status appeared to be predictive of high levels of PTSD symptoms, as shown in previous studies that examined the mental health effects of the SARS outbreak [24, 36, 37].

To improve the mental bell-being of hospital workers during health crises, many authors have suggested that special care should be taken to address the level of anxiety, depression, and PTSD symptoms among both health personnel and public service providers, such as by ensuring clear communication, adequate supplies of protective equipment, and access to psychological intervention [33]. They also propose that during breaks, staff should be provided with food and other daily living needs [16]. It has been noted that concrete measures to develop rest areas/break rooms and the possibility of having leisure and moments of relaxation are more appropriate to addressing the needs of caregivers than formal psychological support [25, 38]. Thus, informal mechanisms may be more successful [39] in which, for example, counsellors or retired nurses visit healthcare workers in rest areas [28].

Future longitudinal research is needed to evaluate the medium- and long-term psychological impact of the pandemic on hospital workers and to identify patterns and the co-occurrence of risk factors for adverse mental health outcomes. Intervention studies in real-world settings should be additionally conducted to investigate under which interventions and specific circumstances resilience may be best fostered and the mental health of frontline professionals supported during and after a disease outbreak.

Our study had several limitations. First, it was based on a single hospital group in Greater Paris, limiting the generalizability of our findings to locations less affected by the pandemic in our country. Second, our survey was conducted after the first wave before knowing that there would be subsequent waves afterwards, which may affect the interpretation of our results. Third, as is common for such surveys, there was probably a selection bias for responders due to the sampling methods. Fourth, the scales we used allowed us to evaluate the level of anxiety, depression, and PTSD symptoms, but cannot alone be used to diagnose these disorders. Finally, the cross-sectional nature of the data also limited our ability to assess whether there could be a causal relationship between the health crisis and mental health outcomes, as no pre-COVID data were available. Indeed, French studies assessing the prevalence of anxiety and depressive symptoms among healthcare workers before the pandemic found high levels, similar to our results, and therefore raise the question of the aggravation of these disorders by the COVID-19 health crisis [40].

Notwithstanding, our study also had several strengths: 1/ the size of our sample, 780 hospital workers, assured that the study was sufficiently powered; 2/ we had a 29% response rate to our survey, which is close to published rates [41–43], especially those published during the COVID-19 pandemic [44–46]; 3/ the homogeneity of the sample, as all respondents were working in the same university hospital in Greater Paris, which was severely affected by the first wave of the pandemic; and 4/ the distribution of our sample is representative of the hospital worker population and included every type of profession, including administrative staff.

## Conclusions

We conducted a cross-sectional survey study to assess socio-emotional factors associated with probable anxiety, depression, and PTSD among 780 hospital workers a few months after the first wave of the COVID-19 pandemic in France.

The level of mental health symptoms found in our study does not appear to be much higher than during the pre-pandemic period. In addition, by including all employees in the survey, it is apparent that the risk of mental suffering for hospital workers was similar, regardless of the professional category (frontline healthcare or non-clinical workers) and whether or not there was direct exposure to ill patients.

However, specific interventions to promote mental well-being in hospital employees exposed to a health crisis such as the COVID-19 pandemic should be implemented as primary prevention. Particular attention should be given to those who are infected or have colleagues or relatives who are infected and those with a history of burnout or depression. Moreover, as secondary prevention, it may be useful to identify hospital workers who feel especially anxious during the crisis, because they appear to be more vulnerable to mental distress afterwards.

Finally, despite the heavy workload and demands that it entails, hospital workers in general are satisfied with their profession, which is a protective factor against psychological pathologies.

## Supporting information

**S1 Appendix. Submitted questionnaire for the survey.**
(DOCX)

**S1 Table. Univariable analysis of factors associated with the presence of emotional symptoms.**
(DOCX)

**S1 Dataset. Full database.**
(XLSX)

## Acknowledgments

The authors would like to thank Jean-Patrick Lajonchère, chief manager of our hospital, for allowing us to conduct this investigation, Alexandra Stulz and Karen Pinot from the post-crisis psychological support work group for their help in discussions and setting up the study, Hélène Beaussier, Nesrine Ben Nasr, Julien Fournier, and Maryline Fleury for their technical and methodological support, Wissem El Hage for having helped us in the development of the questionnaire, and Chloé Lacoste, who agreed to carefully proofread the manuscript for language errors.

## Author Contributions

**Conceptualization:** M. d'Ussel, X. Durand, G. Chatellier, N. Castreau, F. Adam.

**Data curation:** M. d'Ussel, X. Durand, N. Castreau, F. Adam.

**Formal analysis:** M. d'Ussel, A. Fels, X. Durand, G. Chatellier, F. Adam.

**Investigation:** M. d'Ussel, A. Fels, N. Castreau.

**Methodology:** M. d'Ussel, A. Fels, X. Durand, G. Chatellier, F. Adam.

**Project administration:** M. d'Ussel, F. Adam.

**Supervision:** A. Fels, X. Durand, C. Lemogne, F. Adam.

**Validation:** M. d'Ussel, A. Fels, X. Durand, C. Lemogne, G. Chatellier, N. Castreau, F. Adam.

**Visualization:** M. d'Ussel.

**Writing – original draft:** M. d'Ussel, A. Fels.

**Writing – review & editing:** X. Durand, C. Lemogne, G. Chatellier, F. Adam.

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
