## [Decision Letter · Decision Letter 0]

14 Jul 2021

PONE-D-21-17297

Factors associated with psychological symptoms in hospital workers of a French hospital during the COVID-19 pandemic: lessons from the first wave.

PLOS ONE

Dear Dr. d'Ussel,

Thank you for submitting your manuscript to PLOS ONE. After careful consideration, we feel that it has merit but does not fully meet PLOS ONE’s publication criteria as it currently stands. Therefore, we invite you to submit a revised version of the manuscript that addresses the points raised during the review process.

My comments are the following:

With so many predictors an issue of overfitting is a real danger. Can you, please, report on the percentage of variance explained in the DVs (any pseudo R^2^, e.g., Nagelkerke).

What about multicollinearity among the predictors? Was this analysis performed (e.g., Tabachnik and Fidell, 2013)?

Some decisions appear arbitrary or not elaborated properly. For example, why age variable was categorized instead of used as it is? Since you categorized it from the beginning - even in the questionnaire (i.e., it was not done post-hoc) - there must be some reasons for it, that were not elaborated enough. Furthermore, you decided to use the first category (age 18-25) as a referential value. I wonder why, why not the last one? Anyway, you should elaborate on it and justify other decisions in binarizing/categorizing your variables (prof. experience, job satisfaction etc.) 

Generally, why did you decide to use cut off criteria for anxiety, depression, and PTSD and binarize them instead of exploit the full amount of information in these variables as the continual ones? You had rightly mentioned as one of the limitations of your work that these were not proper diagnoses, so why did you choose to behave as they were? Please, elaborate.

In order to make tables more readable I suggest at least to skip unadjusted model or model 1.

We look forward to receiving your revised manuscript.

Kind regards,

Goran Knezevic

Academic Editor

PLOS ONE

Journal Requirements:

"The protocol has been approved by the institutional ethics committee IRB number IRB00012157 (n° initial agreement 436 and registered on national institute of health data platform).

No written informed consent was required. The authors guarantee the anonymization of all data collected. "

3. You indicated that informed consent was not necessary for your study. Could you please provide further details on why your study is exempt from the need for consent and confirmation from your institutional review board or research ethics committee (e.g., in the form of a letter or email correspondence) that ethics review was not necessary for this study? Please include a copy of the correspondence as an "Other" file.

5. Please amend either the title on the online submission form (via Edit Submission) or the title in the manuscript so that they are identical.

6. Please include a copy of Table 2 which you refer to in your text on page 10.

Reviewers' comments:

Reviewer's Responses to Questions

**Comments to the Author**

1. Is the manuscript technically sound, and do the data support the conclusions?

Reviewer #1: Yes

Reviewer #2: Partly

2. Has the statistical analysis been performed appropriately and rigorously? 

Reviewer #1: Yes

Reviewer #2: No

3. Have the authors made all data underlying the findings in their manuscript fully available?

Reviewer #1: Yes

Reviewer #2: Yes

4. Is the manuscript presented in an intelligible fashion and written in standard English?

Reviewer #1: Yes

Reviewer #2: Yes

5. Review Comments to the Author

Reviewer #1: The topisc of the paper is relevant and actual aiming to analyze the mental health problems and emotional status of hospital workers during pandemic of COVID-19. Methods are appropriate, the results and discussion are well done, and limitations of the study is properly recognized.

The Table 1. and 2. are followed by the description which is redundant, so the tables could be deleted or the text should point out only important findings.

Reviewer #2: The manuscript “Factors associated with psychological symptoms in hospital workers of a French hospital during the COVID-19 pandemic: lessons from the first wave” presents data collected during the first wave of the COVID19 pandemic in French hospitals. Findings suggest that healthcare workers during the COVID19 pandemic are at heightened risk for anxiety, depression, and PTSD.

I find the paper timely due to the very heavy burden under which health care professionals are. However, the presented study has several shortcomings, and the majority of my concerns are related to the methodological aspects of the study. In addition, the study is designed to be purely exploratory, and the impact and the relevance of the study is not highlighted at all.

The major problem is the questionnaire used to collect data on socio-demographics. Its structure, type of questions (majority of items are binary and some are multiple responses), and information collected prevent us from making solid judgments and conclusions – this is extremely important as these items are major predictors in the study. Some items (especially those about COVID19 infection) are ambiguous and nothing can be deduced based on the respondents’ answers – were respondents infected or whether they infected someone else? These should have been kept separate to allow for making conclusions. I am missing the rationale why the majority of items are categorical. For example, respondents were grouped by age in several groups, and this also applies to other variables like working experience, patient management, etc. – my advice is to use the variables as continuous instead of categorical variables. Do two hospitals from which participants were recruited differ in some aspects of work or they were selected at random? Is there some reason to believe that healthcare workers would perceive and experience the pandemic differently in these two hospitals?

Importantly, items investigating mental health status before the COVID19 pandemic are not precise, some of them are ambiguous, and we cannot make any solid conclusion about mental health during the pandemic or how prior experience affected current mental health status. Thus, the main aim of the study worded as assessing the psychological impact of the COVID19 pandemic is not feasible at all – the study is cross-sectional and even more, items assessing mental health before the pandemic do not allow us to make some conclusions about it.

I strongly advise changing the analytical strategy. Running so many analyses ended up in having findings compartmentalized, fragmented, and we are missing the big picture about the mental health outcomes in healthcare workers. I suggest running a model in which variables would be included simultaneously to allow for a comprehensible overview of the study variables and their relationships.

Tables are huge and not very user-friendly – try to present data in a more intelligible fashion. Text describing tables is very detailed and redundant.

There are some paragraphs in the discussion section that appear to be disconnected from the rest of the text. For example, the third paragraph on p. 23 “So it’s suggested that concrete measures to develop rest rooms and the possibility of having leisure and moments of relaxation are more appropriate to the needs of caregivers than formal psychological support(22,36), and informal mechanisms might be more successful(37) where, for example, counsellors or retired nurses visit healthcare workers in rest areas(25).” How is this paragraph related to the rest of the discussion?

Language needs proofreading, please check the manuscript carefully – some grammar errors are present in the manuscript.

Overall, due to a large number of issues in the questionnaire used to collect data that I listed above, I cannot recommend this paper for publishing

6. PLOS authors have the option to publish the peer review history of their article (what does this mean?). If published, this will include your full peer review and any attached files.

Reviewer #1: No

Reviewer #2: **Yes: **Ljiljana Lazarevic

---

## [Author Response · Author response to Decision Letter 0]

7 Sep 2021

Response to academic editor: 

We agree with the reviewer. We added the percentage of variance explained (Nagelkerke pseudo R²). 

As you will see on the following tables 1, 2 and 3 there is no over fitting. Moreover we use 2 by 2 tables to test relationship between variables. This lead us to suppress some variables: Professional experience linked with Professional experience in the same hospital unit, and Usual job satisfaction linked with Current job satisfaction.

To elaborate the questionnaire and to decide which variable to include in it, and furthermore how to use these variables (in binarizing/categorizing them), we have studied the previous literature with attention. We have reviewed the literature about the psychological impact of previous epidemics on healthcare workers, and their associated factors. Thus, we have used the different items as they were used in this literature: age (Chong et al. 2004; Wu et al. 2009), work experience (Chong et al. 2004), job title (Chong et al. 2004), marital status(Chong et al. 2004; Lin et al. 2007; Su et al. 2007; 2007; 2007; Wu et al. 2009), past history of psychological disorder (Su et al. 2007), working units (Chong et al. 2004; Su et al. 2007)…

This is always an arbitrary choice. Usually, when a variable is ordered we choose the highest or the lowest category. As regard age we finally binarized the variable, according to the limit corresponding roughly to the median. We also considered only 2 sites (Hospital Paris Saint-Joseph and Hospital Marie-Lannelongue) because most employees working at both hospitals worked mainly in Paris Saint-Joseph. As regard profession, we chose arbitrary as baseline administrative healthcare because those persons belonging to these profession were the less expose to Covid-19.

After discussion with psychiatric experts we have chosen the HAD scale and PCL to determine anxiety, depression and PTSD symptoms even if these scales were not considered as diagnostic scales but as screening tests: the HADS is quick to fill, has a good validity in its French translation, and can be used among any population. According to this review (Bjelland et al. 2002) it must be used by binarization with a cut off. The best advantage with the PCL is that it summarized the characteristics of latest DSM classification for PTSD. This scale is also used with a published binarized cut off (Yao et al. 2003)

In clinical epidemiology it is usual practice to present both adjusted and unadjusted OR. We therefore maintain the unadjusted model in the table. 

When using a hierarchical model it is also the way to present results (Gazmararian et al. 2000)

Table 1 : VIF for each regression logistic for symptoms of anxiety

 Model 1 Model 2

Age 1.07 1.07

Gender 1.03 1.04

Profession 1.11 1.20

Place of professional practice 1.03 1.04

Respondents who were infected or having infected colleagues or relatives 1.02 1.04

Professional burn-out or depression history 1.01 1.04

Anxiety during 1st Wave 1.11

Fear of contaminating relatives during the 1st wave 1.13

Current job satisfaction 1.06

Table 2 : VIF for each regression logistic for symptoms of depression

 Model 1 Model 2

Age 1.06 1.07

Gender 1.02 1.04

Place of professional practice 1.04 1.04

Remote work 1.09 1.09

COVID-Unit assignment 1.13 1.13

Marital situation 1.03 1.03

Respondents who were infected or having infected colleagues or relatives 1.04 1.03

Professional burn-out or depression history 1.02 1.03

Anxiety during 1st Wave 1.03

Current job satisfaction 1.03

Table 3 : VIF for each regression logistic for symptoms of post-traumatic stress

 Model 1 Model 2

Age 1.04 1.05

Gender 1.02 1.03

Place of professional practice 1.05 1.05

Non-clinical professional activity 1.02 1.04

Marital situation 1.03 1.04

Respondents who were infected or having infected colleagues or relatives 1.02 1.03

Professional burn-out or depression history 1.02 1.04

Anxiety during 1st Wave 1.05

Fear of contaminating relatives during the 1st wave 1.05

Current job satisfaction 1.01

Response for Journal Requirements:

The study has been done in accordance with the Declaration of Helsinki and approved by the GERM (Groupe Ethique et Recherche Médicale/ Ethics and Medical Research Group) from the Hospital Paris Saint-Joseph (IRB number 00012157).

This research is part of the institutional care for its employees. In this context, the French regulations (JORF n°0160 of 13 July 2018 text n°110, MR-004) do not require consent but require the transmission of an information note to the employees setting out the purpose of the research. The employees’ non-opposition to the use their data for research purposes is also collected in accordance with the European General Data Protection Regulation (GDPR).

Certification of ethical opinion from the institutional ethics committee is provided as a complementary document.

Data are contained within the Supporting Information files, and available in Excel format on direct request to the first author : mdussel@ghpsj.fr

Response to Reviewer #1:The manuscript has been modified to avoid redundancies, and table 2 has been deleted

Responses to Reviewer#2: To elaborate the questionnaire and choose the variables, we have studied the literature about the psychological impact of previous epidemics on healthcare workers (Chong et al. 2004; Wu et al. 2009; Lin et al. 2007; Su et al. 2007). Especially for the item about COVID-19 infection, we have based on this article (Wu et al. 2009) where it was asked if the healthcare workers had been infected or had friends or relatives who had been infected 

To elaborate the questionnaire and to decide which variable to include in it, and furthermore how to use these variables (in binarizing/categorizing them), we have studied the previous literature with attention. We have reviewed the literature about the psychological impact of previous epidemics on healthcare workers, and their associated factors. Thus, we have used the different items as they were used in this literature: age (Chong et al. 2004; Wu et al. 2009), work experience (Chong et al. 2004), job title (Chong et al. 2004), marital status(Chong et al. 2004; Lin et al. 2007; Su et al. 2007; 2007; 2007; Wu et al. 2009), past history of psychological disorder (Su et al. 2007), working units (Chong et al. 2004; Su et al. 2007)…

Although belonging to the same group, the two hospitals have different specificities, and the recruitment of patients is therefore different. For example, Hospital Marie Lannelongue does not have an emergency department. It was not impacted in the same way during the 1st wave, since 635 patients were hospitalized at Hospital Paris Saint Joseph compared to 199 at Hospital Marie Lannelongue. Therefore, one would have thought that the prevalence of anxiety, depression and PTSD symptoms among employees would be different in the two hospitals.

We agree with this comment, and we notified it in the discussion, line 341: Last, this cross-sectional study cannot demonstrate that COVID-19 is responsible for additional psychological burden in frontline healthcare professionals, as no pre- COVID data are available

As the title of the article indicates, the main objective of the study was to identify factors associated with the mental status of hospital workers. We have modified the last sentence of the introduction to make clear that this objective is the main one. The short title has been also modified

We agree with the reviewer that there are many variables. Thus, as suggested for statistical problem we suppressed some of them and we grouped categories (e.g. age is now binarized as compared with 6 categories in the previous version of the paper). Our strategy was based on the strategy published by (Gazmararian et al. 2000) using a classification variables: demographical, professional and emotional. 

We simplified the tables by regrouping categories for some variables and suppressing redundant ones. We also simplified text to avoid any redundancy between the results chapter and tables. 

Finally, we suppressed table 2.

We agree partly with the reviewer. The text is not directly connected to our results. It is more adapted to a perspective section, where we moved it.

The manuscript has been carefully checked to correct grammar errors

We respect the opinion of the reviewer. However, our goal was to identify factors predictive of mental health impairment at the end of the first wave. We do think that our results obtain with many variables offer perspectives for prevention choices during subsequent waves.

---

## [Decision Letter · Decision Letter 1]

1 Nov 2021

PONE-D-21-17297R1Factors associated with psychological symptoms in hospital workers of a French hospital during the COVID-19 pandemic: lessons from the first wave.PLOS ONE

Dear Dr. d'Ussel,

Thank you for submitting your manuscript to PLOS ONE. After careful consideration, we feel that it has merit but does not fully meet PLOS ONE’s publication criteria as it currently stands. Therefore, we invite you to submit a revised version of the manuscript that addresses the points raised during the review process.

I would suggest you to carefully observe the comments and recommendations of the reviewers, especially R#2 (I will not reiterate these comments). Scientific reports should be simple and easy to understand, sentences more precise, without ambiguities. After removing adjusted coefficients you still preserved the same table descriptions (did not remove word "adjusted"). Please, be careful with such things.  However, the central issue related to your overall findings is the following: you neither obtained higher prevalence of the studied symptoms during Covid19 pandemics (when compared to pre-pandemic conditions, e.g. Hardy et al., 2019), nor you obtained the difference in these symptoms between those working in Covid-19 units and other units. So, in the light of what you have reported, it is closer to the truth that none of the professional categories were affected by the pandemic, than that everyone was affected "not just front-line or even clinical staff" (p. 26).In other words, the relationships that you have obtain between your predictors and symptoms, mostly reveal the usual pattern of symptom correlates independent of the pandemic. Even what appears as something uniquely related to the pandemics - the level of anxiety in the first wave - can reflect nothing more than individual differences in trait anxiety of the participants (especially having in mind that is was measured retrospectively and very roughly with one dichotomous question). You mentioned it sporadically in limitations, but you should make it more salient to a reader. I would recommend to devote full attention to this issue and to elaborate on it properly.   

We look forward to receiving your revised manuscript.

Kind regards,

Goran Knezevic

Academic Editor

PLOS ONE

Reviewers' comments:

Reviewer's Responses to Questions

**Comments to the Author**

1. If the authors have adequately addressed your comments raised in a previous round of review and you feel that this manuscript is now acceptable for publication, you may indicate that here to bypass the “Comments to the Author” section, enter your conflict of interest statement in the “Confidential to Editor” section, and submit your "Accept" recommendation.

Reviewer #1: All comments have been addressed

Reviewer #2: (No Response)

2. Is the manuscript technically sound, and do the data support the conclusions?

Reviewer #1: Yes

Reviewer #2: Yes

3. Has the statistical analysis been performed appropriately and rigorously? 

Reviewer #1: Yes

Reviewer #2: Yes

4. Have the authors made all data underlying the findings in their manuscript fully available?

Reviewer #1: Yes

Reviewer #2: Yes

5. Is the manuscript presented in an intelligible fashion and written in standard English?

Reviewer #1: Yes

Reviewer #2: No

6. Review Comments to the Author

Reviewer #1: The topic of the manuscript is relevant and actual. The authors have adequately addressed r comments of reviewers, so I believe that the manuscript is now acceptable for publication. There still are some minor suggestions:

In the abstract, in 32 line, the authors have written Post-traumatic Stress Distress (PTSD) instead Post-traumatic Stress Disorder

The subtitle Materials and Methods, line 94- It would be recommended to stay only Methods (without Materials)

Reviewer #2: I have reviewed the previous version of the paper “Factors associated with psychological symptoms in hospital workers of a French hospital during the COVID-19 pandemic: lessons from the first wave”. The authors have improved the manuscript, and I would like to thank them for that. However, in their letter with responses they listed responses to the comments without including original comments of reviewers, making it very difficult to say which response correspond to which comment. Therefore, I suggest authors to include both comments of the reviewers and responses in the same file.

Despite manuscript being improved, there are still numerous issues that have to be solved. The most important thing relates to the writing itself – numerous sentences are ambiguous and not clear, several are imprecise, and this must be resolved. This makes manuscript very difficult to read, and impedes making judgements about the results. I’ll state some of the examples:

Abstract – “Significant symptoms of anxiety (41%), depression (21%) and PTSD were reported by 41%, 21% and 14% of the participants, respectively” – not clear why these percentages repeat?

Abstract – “There was no difference regarding type of occupation or assignment in a COVID-19 unit.” – there was no difference in what?

p. 10 – “Compared with female gender, male gender was associated with a significantly lower prevalence of symptoms of anxiety (87% vs. 13%; P=0.001) and PTSD (89% vs. 11%; 178 P=0.029), but a comparable prevalence of depression (83% vs. 17%; P=0.73). “ – check percentages for depression.

p. 10 – “Medical professionals had lower prevalence of anxiety symptoms than other workers (11%, P=0.033).” – lower prevalence than other workers, how are these other workers? Sentence is not clear.

p. 10 – “Healthcare workers not working in a COVID-19 Unit were more prone to depression (60% vs. 40%; P=0.023)” – more prone to depression compared to whom?

p. 16 – “Regarding emotional experience at the time of the first wave, the hospital employees who reported to have been anxious during the first wave had more anxiety (86% vs. 14%; p<0.001)more depression (76% vs. 24%; P<0.001) and more PTSD symptoms (90% vs. 10%; P<0.001) in post crisis than those who have not reported anxiety.” – what is post crisis , sentence is not clear.

p. 16 – “Compared to patients without history 211 of burnout or depression those having such a history had greater anxiety (34% vs. 66%; P<0.001) , depression (61% vs. 39%; P<0.001), and PTSD symptoms (34% vs. 66%; P=0.001).” – please be consistent, either write bigger vs smaller numbers, or smaller vs bigger.

p. 17 – “In multivariable analysis (tables 2, 3 and 4), healthcare workers who reported to have been anxious during the first wave reported more anxiety (OR, 5.83; 95% CI: 3.89-8.90), depression (OR, 1.70; 95% CI: 1.11–2.65) and PTSD symptoms (OR, 4.47; 95% CI: 2.35–9.27)” – it seems that something is wrong with this statement – those who said that are anxious reported they were more anxious – it’s circular, check this.

p. 21 –“ It appears to be a strong relationship 318 between this emotional experience at the time of the first wave and the prevalence of significant symptoms of anxiety, depression and PTSD afterwards.” Perhaps I’m missing something, but this sentence is not clear to me – the paper investigates mental health in the first wave of the pandemic, so I don’t understand to what “significant symptoms of anxiety, depression and PTSD afterwards” refer?

p. 21 – “The fact that introducing this variable in the model resulted in rendering some predictive variables (e.g. gender) non-significant strongly suggests that anxiety at the time of the first wave explains a substantial part of the association between vulnerability factors (e.g. history of depression) and subsequent symptoms of anxiety, depression and PTSD” – again, I don’t understand this sentence. What is the variable introduced in the model? ‘what is the connection between gender, history of depression, and anxiety during the first wave of the pandemics? And how this information informs us about secondary prevention strategies?

The study explores mental health issues after the first wave, but throughout the text authors compare mental health before the first wave, during the first wave and after the first wave. This creates confusion as there are no data to compare mental health during and after the first wave.

Table 1 – number of participants responding to specific items differs, and it would be good to include this information consistently for each item – at the moment, for some items information is given, for some it is omitted.

Table 2 and Table 3 – explain acronyms in the title, HADa and HADd in the notes.

Table 3 – check cell Place of professional practice, duplicated text

Please don’t start sentences with numbers (percentages).

To conclude, the major issue preventing me from recommending the manuscript for publication is the writing which has to be substantially improved.

7. PLOS authors have the option to publish the peer review history of their article (what does this mean?). If published, this will include your full peer review and any attached files.

Reviewer #1: **Yes: **Prof. Tamara Dzamonja Ignjatovic, PhD

Reviewer #2: **Yes: **Ljiljana Lazarevic

---

## [Author Response · Author response to Decision Letter 1]

2 Dec 2021

-Scientific reports should be simple and easy to understand, sentences more precise, without ambiguities. 

+We have revised the entire text to make it more clear, readable and unambiguous

-After removing adjusted coefficients you still preserved the same table descriptions (did not remove word "adjusted"). Please, be careful with such things. 

+We try to make more explicit the statistical methods paragraph. 

We also modified legend of tables 2, 3 and 4.

-However, the central issue related to your overall findings is the following: you neither obtained higher prevalence of the studied symptoms during Covid19 pandemics (when compared to pre-pandemic conditions, e.g. Hardy et al., 2019), nor you obtained the difference in these symptoms between those working in Covid-19 units and other units. So, in the light of what you have reported, it is closer to the truth that none of the professional categories were affected by the pandemic, than that everyone was affected "not just front-line or even clinical staff" (p. 26). +That’s true; our study doesn’t allow us to know if the first wave has impacted the hospital workers emotionally. 

But we can observe that the level of mental suffering is the same whatever the professional category and the proximity of involvement with patients and their families during the 1st wave.

So we have changed the conclusion in order to be more precise.

-In other words, the relationships that you have obtain between your predictors and symptoms, mostly reveal the usual pattern of symptom correlates independent of the pandemic. 

+Exact. And that is interesting according to us! One could have imagined that the frontline workers had higher levels of mental suffering after the 1st wave; our study doesn’t show that.

We have modified the discussion to emphasize that notion 

-Even what appears as something uniquely related to the pandemics - the level of anxiety in the first wave - can reflect nothing more than individual differences in trait anxiety of the participants (especially having in mind that is was measured retrospectively and very roughly with one dichotomous question). You mentioned it sporadically in limitations, but you should make it more salient to a reader. I would recommend to devote full attention to this issue and to elaborate on it properly. 

+We agree with your remark and we added a section about this notion in the discussion:

One factor which was independently associated with higher symptoms of anxiety, depression and PTSD was having had a personal experience of anxiety during the first wave. But it’s difficult to conclude with this observation; the statistical result is interesting and it could be a track of further prospective work to implement strategies of secondary prevention among the most anxious workers during the crisis. On the other hand our retrospectively measure, with a dichotomy question, is not sufficient to conclude about the relation between anxiety during the event and occurrence of distant emotional symptoms.

Comments Reviewer 1

- The topic of the manuscript is relevant and actual. The authors have adequately addressed r comments of reviewers, so I believe that the manuscript is now acceptable for publication. There still are some minor suggestions:

In the abstract, in 32 line, the authors have written Post-traumatic Stress Distress (PTSD) instead Post-traumatic Stress Disorder

+Done

-The subtitle Materials and Methods, line 94- It would be recommended to stay only Methods (without Materials) +Done

Comments Reviewer 2

-I have reviewed the previous version of the paper “Factors associated with psychological symptoms in hospital workers of a French hospital during the COVID-19 pandemic: lessons from the first wave”. The authors have improved the manuscript, and I would like to thank them for that. However, in their letter with responses they listed responses to the comments without including original comments of reviewers, making it very difficult to say which response correspond to which comment. Therefore, I suggest authors to include both comments of the reviewers and responses in the same file.

+We are sorry that you didn’t access to the adequate version of our responses

The file called “point-by-point response to reviewer” included a table with the comments in the colon 1 and our responses in the colon 2.

We will be attentive that you can access to this form for the second rewieving.

-Despite manuscript being improved, there are still numerous issues that have to be solved

The most important thing relates to the writing itself – numerous sentences are ambiguous and not clear, several are imprecise, and this must be resolved

This makes manuscript very difficult to read, and impedes making judgements about the results 

+We edited the manuscript a second time to make it more precise and to avoid any ambiguity.

-Abstract – “Significant symptoms of anxiety (41%), depression (21%) and PTSD were reported by 41%, 21% and 14% of the participants, respectively” – not clear why these percentages repeat? 

+Correction:

Significant symptoms of anxiety, depression and PTSD were reported by 41%, 21% and 14% of the participants, respectively.

-Abstract – “There was no difference regarding type of occupation or assignment in a COVID-19 unit.” – there was no difference in what? 

+There was no difference among workers in anxiety, depression and PTSD symptoms regarding type of occupation or assignment in a COVID-19 unit.

-p. 10 – “Compared with female gender, male gender was associated with a significantly lower prevalence of symptoms of anxiety (87% vs. 13%; P=0.001) and PTSD (89% vs. 11%; 178 P=0.029), but a comparable prevalence of depression (83% vs. 17%; P=0.73). “– check percentages for depression We have verified the figures. There is no mistake in the tables. But inappropriate choice of figures was done in the text of results section. This choice does not change the interpretation of the p-value. 

+We corrected it. 

-p. 10 – “Medical professionals had lower prevalence of anxiety symptoms than other workers (11%, P=0.033).” – lower prevalence than other workers, how are these other workers? Sentence is not clear. We compared the different professional categorizations by regrouping different similar professions in a same group, to limit the number of variables: medical professionals (physician, pharmacologist, biologist), administrative healthcare workers, caregivers (nurse, assistant nurse, nurse manager), others caregivers (Physiotherapist, stretcher-bearer, Radiologic Technologist, Psychologist), midwifes and others workers. 

+The reviewer is right, we clarified the sentence. (line 150, table 1)

-p. 10 – “Healthcare workers not working in a COVID-19 Unit were more prone to depression (60% vs. 40%; P=0.023)” – more prone to depression compared to whom? Tables are correct. Inappropriate choice of figures was done in the text of results section. This choice does not change the interpretation of the p-value. 

+We corrected the results section. 

-p. 16 – “Regarding emotional experience at the time of the first wave, the hospital employees who reported to have been anxious during the first wave had more anxiety (86% vs. 14%; p<0.001)more depression (76% vs. 24%; P<0.001) and more PTSD symptoms (90% vs. 10%; P<0.001) in post crisis than those who have not reported anxiety.” – what is post crisis , sentence is not clear.

+Tables are correct. Inappropriate choice of figures was done in the text of results section. This choice does not change the interpretation of the p-value. We corrected the results section.

-p. 16 – “Compared to patients without history 211 of burnout or depression those having such a history had greater anxiety (34% vs. 66%; P<0.001) , depression (61% vs. 39%; P<0.001), and PTSD symptoms (34% vs. 66%; P=0.001).” – please be consistent, either write bigger vs smaller numbers, or smaller vs bigger 

+Tables are correct. Inappropriate choice of figures was done in the text of results section. This choice does not change the interpretation of the p-value. We corrected the results section. 

-p. 17 – “In multivariable analysis (tables 2, 3 and 4), healthcare workers who reported to have been anxious during the first wave reported more anxiety (OR, 5.83; 95% CI: 3.89-8.90), depression (OR, 1.70; 95% CI: 1.11–2.65) and PTSD symptoms (OR, 4.47; 95% CI: 2.35–9.27)” – it seems that something is wrong with this statement – those who said that are anxious reported they were more anxious – it’s circular, check this. 

+We note that this notion of anxiety during the 1st wave is not clear and we have specified it in the entire document:

In multivariable analysis (tables 2, 3 and 4), healthcare workers who retrospectively reported that they felt anxious during the 1st wave had more symptoms of anxiety (OR, 5.83; 95% CI: 3.89-8.90), depression (OR, 1.70; 95% CI: 1.11–2.65) and PTSD symptoms (OR, 4.47; 95% CI: 2.35–9.27) at the moment of the survey. 

-p. 21 – “It appears to be a strong relationship 318 between this emotional experience at the time of the first wave and the prevalence of significant symptoms of anxiety, depression and PTSD afterwards.” Perhaps I’m missing something, but this sentence is not clear to me – the paper investigates mental health in the first wave of the pandemic, so I don’t understand to what “significant symptoms of anxiety, depression and PTSD afterwards” refer? 

+Our survey had been diffused around 3 months after the end of the 1st wave in our hospital located in Paris, without knowing that other waves would come later.

We wanted to investigate the emotional status of the hospital workers after this event that was the 1st wave.

Our study shows there is a relationship between those who retrospectively felt anxious during the 1st wave, and the prevalence of symptoms of anxiety, depression and PTSD some weeks later:

It appears to be a strong relationship between this anxious experience just at the time of the first wave (reported retrospectively by the responders) and the prevalence of significant symptoms of anxiety, depression and PTSD afterwards.

-p. 21 – “The fact that introducing this variable in the model resulted in rendering some predictive variables (e.g. gender) non-significant strongly suggests that anxiety at the time of the first wave explains a substantial part of the association between vulnerability factors (e.g. history of depression) and subsequent symptoms of anxiety, depression and PTSD” – again, I don’t understand this sentence. What is the variable introduced in the model? ‘what is the connection between gender, history of depression, and anxiety during the first wave of the pandemics? And how this information informs us about secondary prevention strategies?

The study explores mental health issues after the first wave, but throughout the text authors compare mental health before the first wave, during the first wave and after the first wave. This creates confusion as there are no data to compare mental health during and after the first wave. 

+We have totally re-written this section in order to let it more readable:

By using a hierarchical strategy in the multivariable analysis, we observed the role played by the anxious experience on the studied variables: the risk of symptoms of anxiety was higher among women in the model 1, but if we enter the emotional experience as variable in the model 2, symptoms of anxiety and gender were no longer associated. Thus, although woman gender is associated with symptoms of anxiety, the causal pathway for these relation is likely to be through the emotional experience during the crisis.

We found similar results for participants with a history of burnout or depression who had more PTSD symptoms: when we adjusted with the variables of emotional experience during the crisis, history of burnout or depression was no longer related to PTSD symptoms. The relationship between history of burnout or depression and PTSD appears to be mediated by the emotional experience of the crisis.

The impact of the personal experience of anxiety during the crisis was particularly strong, especially when the respondents said that this anxiety was focused on their job. There is a strong relationship between this anxious experience just at the time of the first wave (reported retrospectively by the responders) and the risk of symptoms of anxiety, symptoms of depression and PTSD afterwards. So we can consider that identifying that staffs who feel most anxious during a health crisis and implementing secondary prevention strategies targeted at these staff, we could reduce the prevalence of post-crisis mental health disorders.

-Table 1 – number of participants responding to specific items differs, and it would be good to include this information consistently for each item – at the moment, for some items information is given, for some it is omitted. 

+When the number of participants was 780 (total), we didn’t notified it; but if you think is clearer we notified for each item

-Table 2 and Table 3 – explain acronyms in the title, HADa and HADd in the notes. 

+done

-Table 3 – check cell Place of professional practice, duplicated text

+done

-Please don’t start sentences with numbers (percentages). 

+done

---

## [Decision Letter · Decision Letter 2]

30 Dec 2021

PONE-D-21-17297R2Factors associated with psychological symptoms in hospital workers of a French hospital during the COVID-19 pandemic: lessons from the first wave.PLOS ONE

Dear Dr. d'Ussel,

Thank you for submitting your manuscript to PLOS ONE. After careful consideration, we feel that it has merit but does not fully meet PLOS ONE’s publication criteria as it currently stands. Therefore, we invite you to submit a revised version of the manuscript that addresses the points raised during the review process.

The manuscript has been improved to some extent now. However, there are two major obstacles on the way to its acceptance.

1. I would like you to take suggestion on improvement of the language, by r#2, seriously. It means that you have to have proofread done by the professional translator. Without the substantial improvement of the readability of the text, we will not be able to accept the manuscript. The reason is that, generally speaking, your sentences are ambiguous and, frequently, difficult to understand. 

2. It also seems to me that you did not give due consideration to my comments regarding the way you generally interpreted your results. To reiterate, my understanding is that your overall findings are in accordance with the following scenario: the crisis did not affect the level of symptoms you assessed in health care workers in any substantial manner (namely the prevalence of the symptoms among health care workers did not seem to be higher compared to pre-pandemic conditions); as a consequence, the differences between subgroups of health care workers were not found; relationships between anxiety during the first wave (retrospectively measured) and depression and anxiety could result from the simple fact that people with the higher level of trait anxiety are prone to be more anxious in the past, as well as to the tendency of the people with higher trait anxiety to retrospectively report higher level of anxiety (even if they did not have the level of anxiety they tend to report, which is the limitation of the cross-sectional nature of the study). Obviously, the crucial feature of such a scenario is that it is unrelated to pandemics. You have address this issue in the limitations paragraph (rows 425-430), but I urge you to mention it in conclusions. For example, you can add the first sentence in Conclusions something like...but it does not appear to be elevated significantly compared to the the pre-pandemic conditions. 

We look forward to receiving your revised manuscript.

Kind regards,

Goran Knezevic

Academic Editor

PLOS ONE

Reviewers' comments:

Reviewer's Responses to Questions

**Comments to the Author**

1. If the authors have adequately addressed your comments raised in a previous round of review and you feel that this manuscript is now acceptable for publication, you may indicate that here to bypass the “Comments to the Author” section, enter your conflict of interest statement in the “Confidential to Editor” section, and submit your "Accept" recommendation.

Reviewer #1: All comments have been addressed

Reviewer #2: (No Response)

2. Is the manuscript technically sound, and do the data support the conclusions?

Reviewer #1: Yes

Reviewer #2: Yes

3. Has the statistical analysis been performed appropriately and rigorously? 

Reviewer #1: Yes

Reviewer #2: Yes

4. Have the authors made all data underlying the findings in their manuscript fully available?

Reviewer #1: Yes

Reviewer #2: Yes

5. Is the manuscript presented in an intelligible fashion and written in standard English?

Reviewer #1: Yes

Reviewer #2: No

6. Review Comments to the Author

Reviewer #1: All comments of the reviewers were addressed adequately and carefully considered by the authors. The manuscript have been significantly improved- more clearly written and conclusions are more precisely conducted from the results.

Reviewer #2: I appreciate that the authors tried to improve the manuscript and revise the text. The major drawback of this manuscript is the quality of the language. I understand that the authors tried to improve the manuscript, but it is of absolute importance that the paper be rewritten by a professional (native speaker of the English language). It is not possible to list all problems I noticed, and I flagged only some. If the text is not edited and proofread, I cannot recommend its publication. I understand this feedback is disappointing but my intention is only to make the manuscript better, and I hope that authors will interpret my feedback positively.

Below I list some of the language issues I detected.

Lines 106-107: The entire hospital workers were invited to complete a survey online, with a link sent to the mailing list.

Instead of ENTIRE it should be ALL.

Lines 107-108: This was a questionnaire in "Microsoft forms" electronic format that allows the secure sending and storage of responses.

“THIS” is not clear. I understand what you wanted to say, but it is not grammatically correct. Rewrite the sentence.

Lines 108-109: Inclusion criteria were for the responders to have worked during the period from March 15 to May 15 and to be a volunteer to complete the survey.

This sentence has to be rephrased, wording is odd.

Lines 114-115: They were: (the whole of the submitted questionnaire is accessible as supporting information 115 S1 Appendix).

Again, language is not okay.

Lines 130-131: It was made clear in the questionnaire that the stressful event to which respondents were referring was the health crisis (considered to have been over a month old at the time of the survey).

“over a month old” – not clear to what you refer. This has to be revised.

Lines 170-171: Half of the participants (46%) reported having taken care of patients with COVID-19 frequently (every worked day) or regularly (at least one day a week).

Every working day, not worked day. Also, one day in a week, or once a week.

Lines 171-172: 60% reported that a colleague, a 172 friend, or close relative had been infected by SARS-CoV-2.

The sentence should not start with a number.

Lines 174-176: Among the hospital employees, 62% reported that they were anxious during the period of 1st wave: 86% for their 176 family, 57% for themselves, 51% at work and 42% for the others.

The sentence is not clear. The colon makes it very unclear, how were these percentages obtained. Revise the sentence.

Lines 189-190: This proportion was comparable in the presence or absence of symptoms of depression (17% vs 19%; p=0.73).

I don’t understand this sentence.

Lines 248-254: There were more hospital employees who retrospectively reported at the moment of the survey that the first wave made them anxious in the group with symptoms of anxiety than in the group without symptoms of anxiety (86% vs 46%; p = 0.001), the same was true for symptoms of depression (76% vs 59%; p<0.001). There were more employees who reported a history of burnout or depression in the group with symptoms of anxiety than in the group without symptoms of anxiety (34% vs 15%; p < 0.001), the same was true for symptoms of depression (39% vs 18%; p < 0.001), and PTSD (34% vs 21%; p=0.001).

These sentences are not clear. ? Also, they appear to be circular – more people with anxiety symptoms is in the group with symptom of anxiety. Where there more people with symptoms of depression in the group of people with anxiety or in the group with depression? Second sentence is also not clear to me – what was true for the symptoms of depression and PTSD?

Lines 357-359: But it’s difficult to conclude with this observation; the statistical result is interesting and it could be a track of further prospective work to implement strategies of secondary prevention among the most anxious workers during the crisis.

OBSERVATION is not the appropriate word. Perhaps you can say: We cannot made final conclusion based on results of one study, …

To conclude, it is very difficult to get a complete picture on the quality of the results and discussion when a lot of text is not completely understandable.

Please do your best to improve the language of the text.

7. PLOS authors have the option to publish the peer review history of their article (what does this mean?). If published, this will include your full peer review and any attached files.

Reviewer #1: No

Reviewer #2: **Yes: **Ljiljana Lazarevic

---

## [Author Response · Author response to Decision Letter 2]

4 Feb 2022

1. I would like you to take suggestion on improvement of the language, by r#2, seriously. It means that you have to have proofread done by the professional translator. Without the substantial improvement of the readability of the text, we will not be able to accept the manuscript. The reason is that, generally speaking, your sentences are ambiguous and, frequently, difficult to understand. 

-The manuscript has been entirely corrected by a professional translator.

2. It also seems to me that you did not give due consideration to my comments regarding the way you generally interpreted your results. To reiterate, my understanding is that your overall findings are in accordance with the following scenario: the crisis did not affect the level of symptoms you assessed in health care workers in any substantial manner (namely the prevalence of the symptoms among health care workers did not seem to be higher compared to pre-pandemic conditions); as a consequence, the differences between subgroups of health care workers were not found; Obviously, the crucial feature of such a scenario is that it is unrelated to pandemics. You have address this issue in the limitations paragraph (rows 425-430), but I urge you to mention it in conclusions. For example, you can add the first sentence in Conclusions something like...but it does not appear to be elevated significantly compared to the the pre-pandemic conditions. 

-We modified every sentence in the manuscript which could still suggest that there was a causal relationship between the pandemic and mental health outcomes. 

We added it clearly in conclusion.

Relationships between anxiety during the first wave (retrospectively measured) and depression and anxiety could result from the simple fact that people with the higher level of trait anxiety are prone to be more anxious in the past, as well as to the tendency of the people with higher trait anxiety to retrospectively report higher level of anxiety (even if they did not have the level of anxiety they tend to report, which is the limitation of the cross-sectional nature of the study). 

-That’s true; and we wrote it clearly in discussion (line 312)

The major drawback of this manuscript is the quality of the language. I understand that the authors tried to improve the manuscript, but it is of absolute importance that the paper be rewritten by a professional (native speaker of the English language). It is not possible to list all problems I noticed, and I flagged only some. If the text is not edited and proofread, I cannot recommend its publication. I understand this feedback is disappointing but my intention is only to make the manuscript better, and I hope that authors will interpret my feedback positively.

Below I list some of the language issues I detected. 

-We understood that major obstacle and a professional translator has proofread the entire manuscript. We thank the reviewer for having listed the major issues and suggested corrections

---

## [Editor Report · Decision Letter 3]

1 Apr 2022

Factors associated with psychological symptoms in hospital workers of a French hospital during the COVID-19 pandemic: lessons from the first wave.

PONE-D-21-17297R3

Dear Dr. d'Ussel,

We’re pleased to inform you that your manuscript has been judged scientifically suitable for publication and will be formally accepted for publication once it meets all outstanding technical requirements.

Kind regards,

Goran Knezevic

Academic Editor

PLOS ONE
---

## [Editor Report · Acceptance letter]

20 Apr 2022

PONE-D-21-17297R3 

Factors associated with psychological symptoms in hospital workers of a French hospital during the COVID-19 pandemic: lessons from the first wave 

Dear Dr. d'Ussel:

I'm pleased to inform you that your manuscript has been deemed suitable for publication in PLOS ONE. Congratulations! Your manuscript is now with our production department. 

Kind regards, 

on behalf of

Prof Goran Knežević 

Academic Editor

PLOS ONE